

# Impact of uncertainties in inorganic chemical rate constants on tropospheric composition and ozone radiative forcing

Ben Newsome[1] and Mat Evans[1,2]

[1]Wolfson Atmospheric Chemistry Laboratories, Department of Chemistry, University of York, York, YO10 5DD, UK.
[2]National Centre for Atmospheric Science, Department of Chemistry, University of York, York, YO10 5DD, UK.

*Correspondence to:* Mat Evans (Mat.Evans@york.ac.uk)

**Abstract.** Chemical rate constants determine the composition of the atmosphere and how this composition has changed over time. They are central to our understanding of climate change and air quality degradation. Atmospheric chemistry models, whether online or offline, box, regional or global use these rate constants. Expert panels synthesise laboratory measurements, making recommendations for the rate constants that should be used. This results in very similar or identical rate constants being used by all models. The inherent uncertainties in these recommendations are, in general, therefore ignored. We explore the impact of these uncertainties on the composition of the troposphere using the GEOS-Chem chemistry transport model. Based on the JPL and IUPAC evaluations we assess 50 mainly inorganic rate constants and 10 photolysis rates, through simulations where we increase the rate of the reactions to the $1\,\sigma$ upper value recommended by the expert panels.

We assess the impact on 4 standard metrics: annual mean tropospheric ozone burden, surface ozone and tropospheric OH concentrations, and tropospheric methane lifetime. Uncertainty in the rate constants for $NO_2 + OH \xrightarrow{M} HNO_3$, $OH + CH_4 \rightarrow CH_3O_2 + H_2O$ and $O_3 + NO \rightarrow NO_2 + O_2$ are the three largest source of uncertainty in these metrics. We investigate two methods of assessing these uncertainties, addition in quadrature and a Monte Carlo approach, and conclude they give similar outcomes. Combining the uncertainties across the 60 reactions, gives overall uncertainties on the annual mean tropospheric ozone burden, surface ozone and tropospheric OH concentrations, and tropospheric methane lifetime of 11, 12, 17 and 17% respectively. These are larger than the spread between models in recent model inter-comparisons. Remote regions such as the tropics, poles, and upper troposphere are most uncertain. This chemical uncertainty is sufficiently large to suggest that rate constant uncertainty should be considered when model results disagree with measurement.

Calculations for the pre-industrial allow a tropospheric ozone radiative forcing to be calculated of $0.412 \pm 0.062$ $Wm^{-2}$. This uncertainty (15 %) is comparable to the inter-model spread in ozone radiative forcing found in previous model-model inter-comparison studies where the rate constants used in the models are all identical or very similar. Thus the uncertainty of tropospheric ozone radiative forcing should expanded to include this additional source of uncertainty. These rate constant uncertainties are significant and suggest that refinement of supposedly well known chemical rate constants should be considered alongside other improvements to enhance our understanding of atmospheric processes.





## 1 Introduction

The concentration of gases and aerosols in the atmosphere have changed over the last century due to human activity. This has resulted in a change in climate (Stocker, 2014) and a degradation in air quality (Dockery et al., 1993) with tropospheric ozone ($O_3$) and methane ($CH_4$) playing a central role. The response of these compounds to the changing emissions is complex and non-linear (Lin et al., 1988). The hydroxyl radical (OH) plays a central role in this chemistry as it initiates the destruction of many pollutants (notably $CH_4$) and so determines their lifetime in the atmosphere. The dominant source of OH is the photolysis of $O_3$ in the presence of water vapour. The oxidation of compounds such as $CH_4$, carbon monoxide (CO) and other hydrocarbons can lead to the production of $O_3$ if sufficient oxides of nitrogen ($NO_x$) are present. Changes in the emissions of $O_3$ precursors between the pre-industrial ($\sim$1850) and the present day have increased $O_3$ concentrations and this has produced a radiative forcing estimated to be $410 \pm 65$ mWm$^{-2}$ (Stevenson et al., 2013).

The rate constants of the reactions occurring in the atmosphere have been determined by a number of laboratory studies which are synthesised by groups such as the IUPAC (Atkinson et al., 2004) and JPL (Sander et al., 2011) panels. These provide recommendations for both rate constants and their associated uncertainties. These reactions are typically expressed in an Arrhenius form to represent the temperature dependence. More complicated representations are needed for three-body reactions. The $1\sigma$ uncertainty in a rate constant at a temperature (T) is expressed as an uncertainty at 298K (f (298)) together with a term (g) that expresses how quickly the uncertainty increases away from 298K (Equation 1), leading to temperature dependences which increase away from room temperature (Figure 1).

$$f(\mathrm{T}) = \mathrm{f}(298\mathrm{K}) \exp \left| g \left( \frac{1}{\mathrm{T}} - \frac{1}{298\mathrm{K}} \right) \right| \tag{1}$$

For the reactions studied, the uncertainty at 298K typically ranges from 5% for well understood reactions to 30% for those which have significant uncertainty. Other reactions can have larger uncertanties then quoted here. The increase in uncertainty at temperatures away from 298K can range from 0% to over 40%, giving some reactions a total uncertainty of over 50% in the cold upper troposphere.

Models of atmospheric composition (whether online or offline, single box or transport etc.) use these recommended rate constants, together with estimates of the meteorology, emissions, deposition, photolysis, etc. of compounds to calculate the concentration of species in the atmosphere. These models are a central tool for our understanding of atmospheric processes and for making policy choices to minimize climate change and air pollution.

Although these models have been developed significantly over the last decades, they have, in general, all used the same basic chemical rate constants as evaluated by the IUPAC or JPL panels. Little emphasis has been placed on understanding the uncertainty in predicted atmospheric composition caused by the uncertainty in these rate constants. The focus has been to investigate the impacts of novel chemical reactions, understanding emissions etc. (e.g. (Sherwen et al., 2016; Hartley and Prinn, 1993)). Here though, we investigate the impact of this uncertainty on the composition of the troposphere. We base our assessment on the uncertainties in rate constants described by the JPL and IUPAC panels (Sander et al., 2011; Atkinson et al., 2004) using the GEOS-Chem model and evaluate a range of model diagnostics for both the present day and the pre-industrial.





## 2   Model simulations

GEOS-Chem (Bey et al., 2001) (www.goes-chem.org) is an offline chemistry transport model. We use version v9-2. For computational expediency we use a horizontal resolution of $4^{°}$ latitude by $5^{°}$ longitude with 47 vertical hybrid pressure-sigma levels from the surface to 0.01 hPa. The chemistry is solved within the troposphere with the SMV-Gear solver (Jacobson and Turco, 1994). We use a mass based scheme for aerosol (Park et al., 2003) and so can not investigate the impact of the rate constant uncertainty on aerosol number or size distribution. Stratospheric chemistry is unchanged in all simulations and uses a linearised approach to the chemistry (McLinden et al., 2000; Murray et al., 2012). Global anthropogenic emissions were taken from the Emission Database for Global Atmospheric Research (EDGAR) v3 for NOx, CO,VOCs and SOx. Regional or source specific inventories replaced EDGAR where appropriate (EMEP, BRAVO, Streets, CAC, NEI05, RETRO, AEIC see the GEOS-Chem wiki for more details). Biogenic emissions (Isoprene, Monoterpenes, Methyl Butenol) are taken from the MEGAN v2.1 emission inventory (Sindelarova et al., 2014). Biomass burning emissions were used from the GFED3 monthly emission inventory(van der Werf et al., 2010). $NO_x$ sources from lightning (Murray et al., 2012) and soils (Hudman et al., 2012) were also included. As in previous studies (Parrella et al., 2012; Sofen et al., 2011) pre-industrial emissions are calculated by switching off anthropogenic emissions, reducing biomass burning emissions to 10% of their modern day values, and by setting $CH_4$ concentrations to a constant 700 ppbv (Parrella et al., 2012).

For both present-day and the pre-industrial simulations we run the model from the 1st of July 2005 to the 1st of July 2007 with GEOS-5 meteorology. We used the first year to spin up the composition of the troposphere. Metrics are derived from the second year of simulation.

We follow the methodology of JPL (Sander et al., 2011) for the representation of uncertainties in rate constants. For two body reactions the uncertainty is given by two parameters. f (298K) describes the relative uncertainty at 298K, and g describes how the uncertainty increases as temperature diverges from 298K, as shown in equation (1).

## 3   Reactions Studied

We limit our study to the inorganic ($O_x$, $HO_x$, $NO_x$, CO, $CH_4$) reactions together with some key organic and sulfur reactions. Mechanistic uncertainties in the organic chemistry of the atmosphere makes a systematic assessment of these uncertainties difficult (Goldstein and Galbally, 2007). Table 1 shows a list of reactions that are perturbed and the uncertainties assumed. We use the uncertainty recommendations from the JPL panel if provided and the IUPAC panel otherwise. We investigate the impact of 50 inorganic chemical reactions and 10 photolysis reactions (Table 1). Uncertainties in photolysis rate constants are harder to define than for the other reactions. We consider the appropriate chemical uncertainty here as the uncertainty in the absorption cross section and the quantum yield rather than the uncertainty in the photon flux which we attribute to the radiative transfer calculation. A full calculation of the chemical uncertainty in a photolysis rate is complex as it it depends upon the uncertainties at different wavelengths, the independence of the cross section and quantum yield parameters and the transfer of this information through the spectral bins used for the laboratory studies and the photolysis calculations. In order to simplify this calculation we apply a 10% uncertainty to all photolysis rates.





## 4   Single Reaction Perturbations

From each of these 60 reactions we increase the reaction rate by the 1 $\sigma$ temperature dependent uncertainty given in Table 1. To

allow the model to spin up we take the 2nd year of simulation and calculate four metrics: tropospheric $O_3$ burden, mean surface $O_3$ mixing ratio, tropospheric mass weighted mean OH number density, and tropospheric mean $CH_4$ lifetime. We subtract the values of these metrics from the base value of the metric (unchanged rate constants) and then take the absolute value to remove cases where the value decreases on an increase in the rate constant. Figure 2 shows the changes for all four metrics with Table 1 giving the values for the change in tropospheric $O_3$ burden. We express these values as a percentage of the base case value.

It is evident that a relatively small number of reactions produce large uncertainties in the values of these metrics. The one that offers the most uncertainty is the reaction between $NO_2$ and OH to product nitric acid which leads to uncertainties in the range of 6–11% in the metrics investigated here. This reaction is both highly uncertain (f (298K)=30%) and acts as a large global sink for $NO_x$ and $HO_x$. The next most significant reaction is that between $CH_4$ and OH to produce $CH_3O_2$ radicals. The model assumes a constant $CH_4$ concentration so an increase in the rate constant between $CH_4$ and OH leads to an increased source

of radicals but doesn't lead to a commensurate drop in the $CH_4$ concentration. Thus an increase in this rate constant in the model is effectively the same as an increase in the emission of $CH_4$ which results in a wide range of impacts such as increased CO concentrations etc. The $O_3$+NO reaction to produce $NO_2$ is central to the partitioning of $NO_x$ in the atmosphere. Thus increasing its rate constant reduces NO concentrations in the atmosphere (leading to lower $O_3$ concentrations) and increasing the concentration of $NO_2$ (which favours $NO_2$ removal) which again reduces $O_3$ concentrations. The tenth most significant

reaction for all the metrics generates an uncertainty of less than 1%.

The relative importance of the different reactions does not change much with the metric being investigated (see Figure 2). The rate constants of these top ten reactions are not particularly uncertain (other than for $NO_2$+OH) compared to other reactions but they link important chemical cycles and have a very large chemical flux flowing through them. Thus relatively small changes in their uncertainties will lead to large changes in concentration.

Given the uncertainties for the individual reactions calculated here, the next question is as to how these uncertainties can be combined together to generate a single uncertainty from rate constants uncertainty on the composition of the atmosphere.

## 5   Addition of uncertainties

If these perturbations are independent (uncertainties in one rate constant are not related to uncertainties in another) and the model approximately linear, the total rate constant uncertainty can be found by finding the root of the sum of the individual uncertainties squared (addition in quadrature) as shown in equation (2).

$$\sigma^2_{total} = \Sigma \sigma^2_{reaction} \tag{2}$$





It is hard to assess the independence of the rate constants. Given the nature of the laboratory experiments used to determine them, it is likely that there is some overlap in assumptions. It would be extremely difficult to diagnose this for all 60 reactions and so we ignore this in further work.

Atmospheric chemistry is non-linear (Lin et al., 1988). A doubling of a change to the model, does not necessarily lead to a doubling of the model response. Thus, is it not obvious how uncertainties from the individual rate-constant perturbations should be combined. To investigate this we perform a Monte Carlo analysis of the model. We take ten of the most significant reactions determined earlier (shown by the * in Table 1) and generate 10 normally distributed random numbers ($\mu = 0$, $\sigma=1$), one for each reaction. For each of the ten rate constants we add on the calculated $1\sigma$ uncertainty multiplied by the random number and run the model. We repeat this 50 times to produce a Monte-Carlo ensemble from which we can calculate the four metrics described earlier.

If the model is linear, the metrics calculated from each member of the Monte Carlo ensemble should be (to some level) the same as the linear addition of the individual rate-constant perturbations weighted by the Monte Carlo random numbers. Figure 3 shows the perturbation in the value of the metric calculated for each ensemble member against the calculated value of the metric using the single reaction values. The model shows a strong linear relationship between the metrics examined (intercepts of $0.21\pm0.9$ % and gradients of $0.80\pm0.04$) thus if the errors are uncorrelated we can, at least to a first approximation, add the individual $1\sigma$ perturbations together in quadrature using Equation 2 to calculate the overall uncertainty in the model metrics. From these simulations we estimate the quadrature approach leads to an over-estimate of the $1\sigma$ uncertainty on the order of 10%.

We thus conclude that the adding together of the individual perturbations in quadrature gives a good approximation to the uncertainty calculated by the Monte Carlo method for significantly less computational burden.

## 6 Impacts on the present day atmosphere metrics

We show on Figure 2 the absolute percentage change in global annual mean $O_3$ burden, surface $O_3$, tropospheric average OH and $CH_4$ tropospheric lifetime from increasing each of the reaction rate constants in Table 1 in turn by their $1\sigma$ value. They are ordered by the magnitude of the perturbation and for clarity we only show the top 20, combining the remaining 40 in quadrature into the 'Other' category. The fractional change in tropospheric $O_3$ burden for all of the perturbations is given in Table 1. We show the results of combining all of these reactions in quadrature ('Total (sum)'), the result of combining the top 10 in quadrature ('Top 10') and the standard deviation from the 50 Monte-Carlo simulations ('Monte Carlo Top 10'). The relative closeness (~ 10%) of the value calculated from the 'Top 10' and the 'Monte Carlo Top 10' shows that the addition in quadrature approach provides a useful approximation to the Monte Carlo methodology with significantly less computational burden.

The top ten reactions contribute over 90% of the uncertainty for all metrics with the overall uncertainty for the annual mean tropospheric ozone burden, surface ozone and tropospheric OH concentrations, and tropospheric methane lifetime of calculated to be 11, 12, 17 and 17% respectively. These uncertainties can be compared to the inter-model spreads found from model inter-





comparison exercises. The multi-model standard deviation in the ozone burden, tropospheric OH concentration and troposphere methane lifetime were found to be 7%, 10% and 10% in the ACCMIP studies (Young et al., 2013; Voulgarakis et al., 2013).

Thus we find that the chemical rate constant uncertainty is larger than the multi-model spread which is usually used to give some sense of our uncertainty in our understanding of a quantity. As the models used in these inter-comparisons typically use the same rate constants, this rate constant uncertainty is not included in the inter-model spread and so the inter-model spread should be considered the lower estimate for the uncertainty on parameters.

## 7  Spatial distribution of uncertainty

Figure 4 shows the spatial distribution of the total uncertainty in the annual mean $O_3$, OH and CO concentration, for the tropospheric column, the zonal mean, and at the surface from the 60 reactions. Similar plots for a large number of other model species are shown in Figures 5–10. There is a significant degree of in-homogeneity in these uncertainties which respond to a range of factors. The uncertainties in the rate constants are largest in the upper troposphere where the temperatures are coldest and thus furthest from the 298K base temperature used to calculate the uncertainties. However, these uncertainty can only

manifest if chemistry is the large source or sink for a species in that region. $O_3$ uncertainties are relatively low in the upper troposphere as it has a large stratospheric source in this region which we have not perturbed (see Section 2). OH uncertainties on the other hand are high ( 30%) in the upper troposphere due to the low temperatures. Over continental regions the concentration of CO is not particularly uncertain as the emissions and transport control the concentration. However, over the ocean where emissions are small, the chemistry becomes more important and so uncertainty increases. Uncertainties in the CO are largest in

the southern hemisphere where direct emission is low and chemical production from $CH_4$ and other hydrocarbons is significant. In general uncertainties are largest over remote regions far from recent emissions, especially if they are particularly cold or hot compared to room temperature. Thus surface OH values are more uncertain in the cold remote southern ocean than they are in the tropics. Surface $O_3$ values are uncertain in the warm tropics where intense sunlight and high water vapour concentrations leads to a large chemical flux through $O_3$.

Across the full set of simulated compounds (Figures 5–10) there are even larger uncertainties. For primary emitted hydrocarbons, large uncertainties occur in remote, photochemically active locations such as the topics where shorter lived hydrocarbons may be many OH lifetimes away from sources. Uncertainties in the OH concentrations thus multiply in these regions, leading to uncertainties of up to 60% for $\geq$C4 alkanes. Secondary products such as $H_2O_2$, $CH_3OOH$ also show significant uncertainties of up to 56% in some locations.

$NO_x$ concentrations close to emission sources are dominated by the emission and transport and so are not very sensitive to chemical uncertainty (Figure 7). However, away from these emissions uncertainties can build up. Uncertainty in the $NO_x$ concentrations at the poles are up to a factor of 40%. Uncertainties in PAN concentrations 8 are in general high (>20%) in most locations ($\sim$ 50% over the remote ocean) reflecting the complexity of the chemistry involving uncertainties in both $RO_x$ and $NO_x$ concentrations. Uncertainties in nitric acid (the dominant $NO_x$ sink) concentrations are smaller however ($\sim$5%) reflecting





the mass balance constraint of emissions of $NO_x$ having to balance $NO_y$ sinks. Large variability in nitric acid concentrations in the southern ocean reflects non-linearities in aerosol thermodynamics of $HNO_3$ / $NO_3^-$ partitioning.

$SO_2$ concentrations show the largest uncertainties in the tropical upper troposphere where OH is also highly uncertain. However, $SO_4^{2-}$ shows much smaller uncertainty, again reflecting mass conservation constraints. $NH_4$ concentrations show little sensitivity to the rate constants analysed. Overall this suggests that aerosol mass is not particularly sensitive to the gas phase chemistry examined here.

Overall, we see a complex pattern of uncertainty with geographically highly variable uncertainty.

## 8   Implications for model-measurement comparisons

Comparisons between the predictions made by models and observations underpin the assessment of model fidelity. Deviations between model and measurements are often used to diagnose model failings. Attributing these differences to uncertainties in the emissions is particularly popular (see for example Hartley and Prinn (1993); Huang et al. (2008)). Figure 11 shows observed monthly mean and standard deviations for CO, $O_3$, $C_2H_6$, $C_3H_8$, $C_4H_{10}$ and $NO_2$ from the World Meteorological Organisation's Global Atmosphere Watch Cape Verde Atmospheric Observatory (Carpenter et al., 2011), overlaid with the base model simulation and the chemical uncertainty ($1\sigma$) calculated from the addition in quadrature of the 60 $1\sigma$ simulations. We chose this location as it is far from recent emissions and so should show large uncertainties for primary emitted species.

Consistent with Figures 5–10 the uncertainty in the model calculation ranges from 5–30% depending upon the species. For some of the species (CO, $O_3$, $C_2H_6$, $C_4H_{10}$) much of the difference between the model and the measurements lie within the model $1\sigma$ uncertainty. For others such as $C_3H_8$ or $NO_2$ the differences are harder to explain and other processes (emissions, transport, unknown chemistry etc.) would need to be explored.

Figures 5–10 show significant changes in uncertainty with changes in the vertical due to increasing uncertainty with reducing temperature. Figure 13 shows a selection of ozonesonde observations from the World Ozone and Ultraviolet Data Centre (WOUDC) compared to equivalent modelled concentrations and uncertainties. Observations are derived from the surface into the middle troposphere as the temperature drops. The uncertainty thus maximises at around 10km. Above this much of the ozone in the model is produced in the stratosphere which is unperturbed in these simulations. Above this height the uncertainty in the ozone due to tropospheric chemistry uncertainty reduces.

These comparisons with observations highlight the complexity of attributing model failure to a particular cause. For some locations and for some species the chemical uncertainty can be large. For the same species, in a different location, the uncertainties may be much smaller. Inversion studies which attempt to attribute model failure to a single cause (for example uncertainties in emissions) need to have a detailed understanding of the magnitude and geographical distribution of the other model errors. We show here that they vary between different species, can be large and highly spatially varying. This should be considered when model inversion studies are undertaken.



## 9 Ozone radiative forcing

We repeat the 60 $1\sigma$ simulations described above with pre-industrial (notionally the year 1850) emissions (see Section 2) to allow us to calculate an uncertainty in the radiative forcing of $O_3$. For each reaction we calculate the difference in the annual mean tropospheric column $O_3$ (Dobson Units) between the present day and pre-industrial with the rate constant increased to its $1\sigma$ value. Then using a linear relationship between change in $O_3$ column and radiative forcing (Stevenson et al., 2013; Young et al., 2013) of $42 mW\, m^{-2}\, DU^{-1}$, we calculate a radiative forcing associated with the uncertainty associated with each reaction. We estimate an overall uncertainty in the tropospheric $O_3$ radiative forcing in the same way as the other metrics, by adding them together in quadrature. In our base simulations we calculated the tropospheric $O_3$ radiative forcing to be $412$ $mWm^{-2}$ consistent with previous studies ($410\pm65 mWm^{-2}$) (Stevenson et al., 2013). Our estimate of the uncertainty in the $O_3$ radiative forcing from rate constant uncertainty is $56\ mWm^{-2}$ (15%) with reaction specific detail shown in Figure 14. Again the same set of reactions contribute the largest share to the uncertainty in the radiative forcing as in the uncertainty in present day $O_3$ burden.

This uncertainty estimate of 15% can be compared to the 17% spread in the $O_3$ radiative forcing calculated between climate models in the recent ACCMIP (Young et al., 2013) inter-comparison (shown in Figure 14). This spread is usually used as the uncertainty in our understanding of $O_3$ radiative forcing. However, as all of these models use the same JPL or IUPAC recommended rate constants the inter-model spread does not include the rate constant uncertainty explored here. Given that the rate constant uncertainty is comparable to the inter-model spread, it should be included in future assessment of the uncertainty in $O_3$ radiative forcing. A naive addition in quadrature approach would suggest that the uncertainty on tropospheric $O_3$ radiative forcing should be increased by roughly 30% to account for this.

## 10 Discussion

We have shown that the uncertainty in the inorganic rate constants leads to significant ($>10\%$) uncertainties in the concentration of policy relevant metrics of troposphere composition ($O_3$ burden, surface $O_3$, global mean OH, tropospheric $CH_4$ lifetime, $O_3$ radiative forcing) with significantly higher uncertainty in other compounds. This uncertainty may have implications for climate policy through an underestimate of the uncertainty on $O_3$ radiative forcing or significant uncertainties on the $CH_4$ lifetime. This also has implication for how model-measurement disagreements are interpreted. Similar conclusions have been found for regional air quality focussed models (Yang et al., 2000).

The simulation performed here likely provide a lower limit to the chemical uncertainty. We do not explore the impact in uncertainties in organic chemistry (beyond that from the initiation of hydrocarbon oxidation) or in organic mechanisms; we do not included tropospheric bromine, iodine, chlorine chemistry in our analysis or heterogeneous parameters. We have neither investigated the impact of rate constant uncertainty on the composition of the stratosphere or mesosphere, or how this may propagate through to the troposphere. There are also uncertainties in the Henry's Law constants used for wet and dry parameterisations etc. It seems likely therefore that the true chemical uncertainty in the composition of the atmosphere is significantly higher than that found here.



5     Although it may be challenging, reducing these uncertainties would provide significant benefits. Targeting the top 10 re-
actions identified here (Figure 2 (a)) would significantly reduce the overall chemical uncertainties. Despite the fact that these
reactions may appear rather un-interesting to some, they provide the basis for determining the composition of the atmosphere.
Given the difficulties in reducing the uncertainties in other areas of the climate system (we will never know the pre-industrial
emissions well etc.) a redoubled effort to reduce rate constant uncertainty appears to be a relatively straightforward methodol-
10   ogy to improve our understanding of atmospheric composition.

*Acknowledgements.* Ben Newsome was supported by a NERC Studentship (NE/L501761/1). This work was supported by the NERC
funded BACCHUS project (NE/L01291X/1). The Cape Verde Atmospheric Observatory is supported by the NERC funded ORC3 project
(NE/K004980/1) and by the National Centre for Atmospheric Science. GEOS-Chem (www.geos-chem.org) is a community effort and we
wish to thank all involved in the development of the model. We would also thank all the JPL and IUPAC panels for their efforts in compiling
atmospheric rate constants.





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





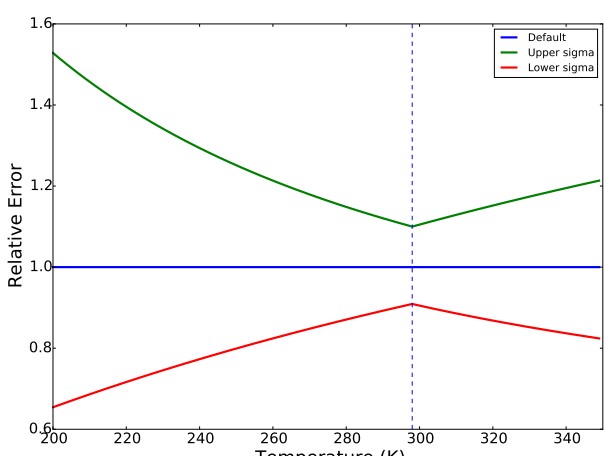

**Figure 1. Example of the uncertainty on a reaction rate constant.** The relative uncertainty of the reaction $O_3$ + NO is plotted as a function of temperature. The lowest uncertainty is at room temperature (298K) with exponentially increasing uncertainties occurring as we diverge to higher and lower temperatures.




**Figure 2. Uncertainties in all metrics.** Fractional uncertainties of **a** $O_3$ tropospheric burden, **b** OH tropospheric burden, **c** $O_3$ surface concentration and **d** $CH_4$ lifetime. Each bar labelled with a reaction represents a run with a $1\sigma$ increase in the rate constant. 'Other' represents the addition in quadrature of the reactions that were not the top 20 most influential. 'Total (Top 10)' represents the addition in quadrature of the 10 most important reactions, and 'Monte Carlo Top 10' represents the standard deviation of the Monte Carlo ensemble. 'Total' represents the addition in quadrature of all the simulations.







**Figure 3. Monte Carlo simulations to understand the models linearity.** The X axis values shows the percentage change in the metric value of an ensemble member compared to the simulation with no perturbations. The Y axis values show the expected percentage change of the metric based on a linear addition of the individual 1 sigma perturbation experiments weighted by the Monte Carlo perturbation values. Metrics investigates are **a** $O_3$ tropospheric burden, **b** $O_3$ mean surface concentration, **c** OH tropospheric burden and **d** $CH_4$ lifetime. We show the result of 50 Monte Carlo simulations. Each simulation perturbs 10 of the most important reactions (* reactions in SI Table 1) $1\sigma$ by normally distributed random numbers.




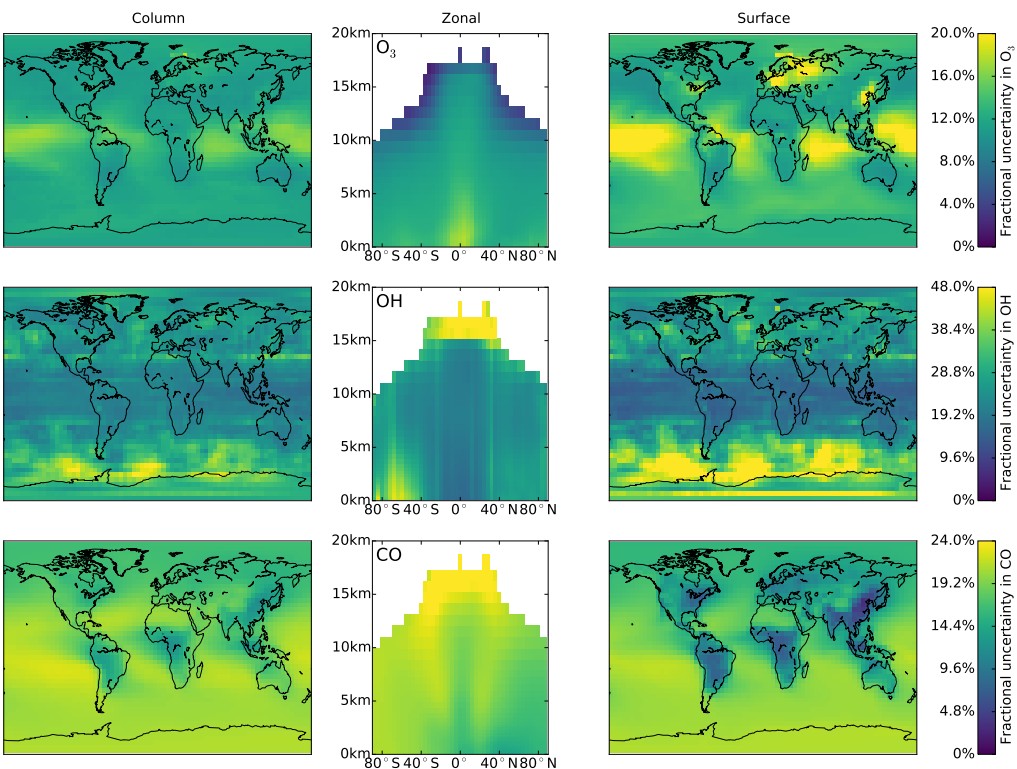

**Figure 4. Spatial distribution of uncertainties.** Fractional uncertainties calculated for $O_3$, OH and CO concentrations for the tropospheric column (left), the zonal mean (centre) and the surface (right) from adding together the individual reaction uncertainties from the 60 reactions studied in quadrature



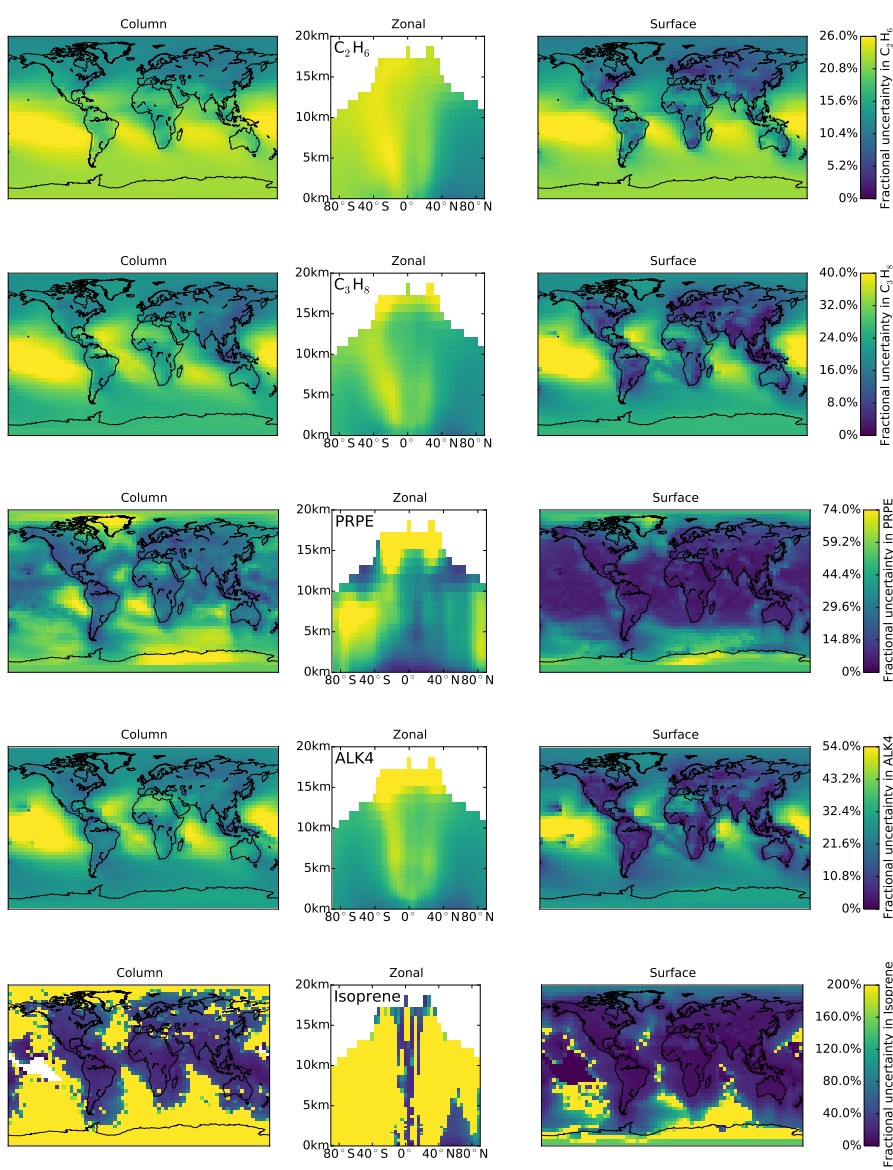

**Figure 5. Primary VOCs.** Total $1\sigma$ uncertainty in the concentrations of $C_2H_6$, $C_3H_8$, PRPE ($\geq$ C3 Alkenes), ALK4 ($\geq$ C4 Alkanes) and ISOP (Isoprene) from the addition in quadrature of the individual reaction uncertainties. Column covers the tropospheric column.







**Figure 6. Other Organics.** Total $1\sigma$ uncertainty in the concentrations of $CH_2O$, MP (Methyl Hydro Peroxide), ALD2 (Acetaldehyde), GLYC (Glycoaldehyde), MACR (Methacrolein) and MKV (Methyl Vinyl Ketone) from the addition in quadrature of the individual reaction uncertainties. Column covers the tropospheric column.



**Figure 7. NO$_x$.** Total $1\sigma$ uncertainty in the concentrations of NO, NO$_2$, NO$_3$, N$_2$O$_5$, HNO$_2$ and HNO$_4$ from the addition in quadrature of the individual reaction uncertainties. Column covers the tropospheric column.



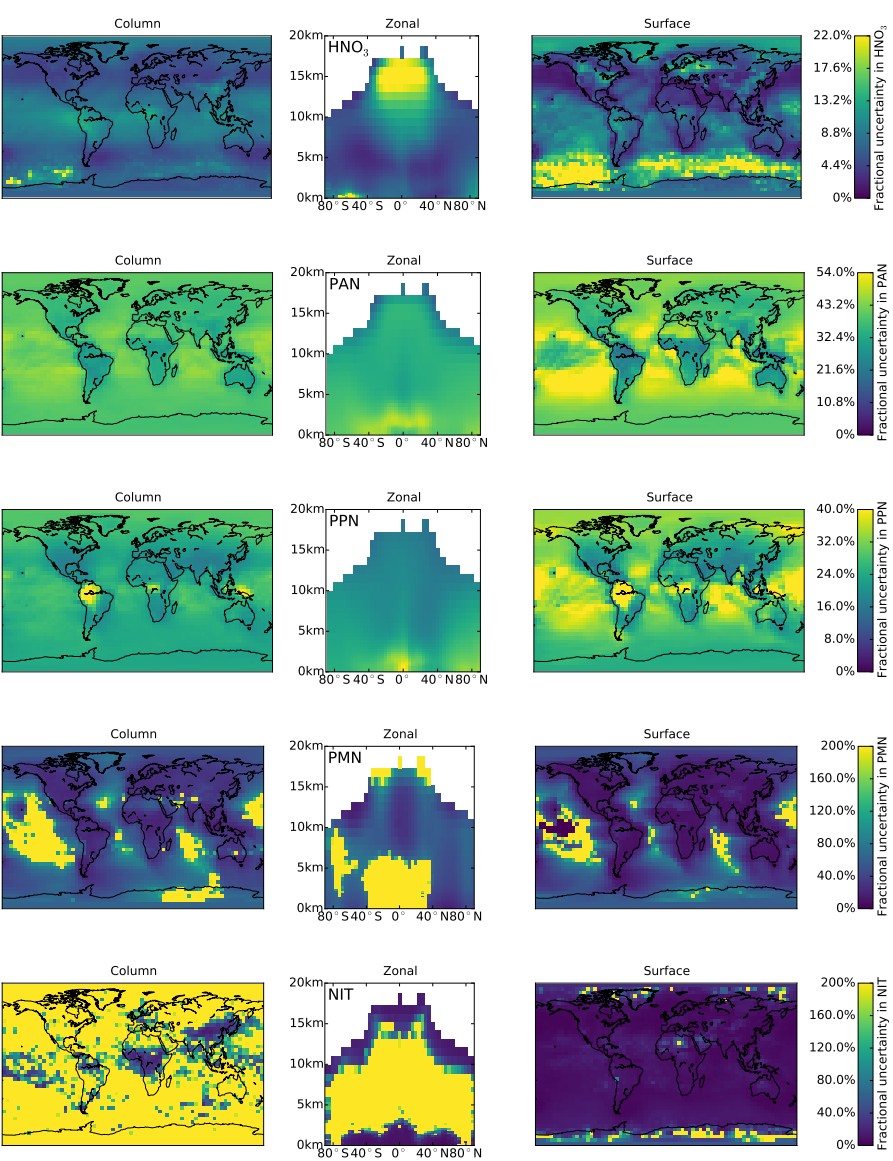

**Figure 8. NO$_y$.** Total $1\sigma$ uncertainty in the concentrations of HNO$_3$, PAN (Peroxyacetyl Nitrate), PPN (Peroxymethacroyl Nitrate), PMN (Peroxymethacroyl Nitrate) and NIT (Inorganic nitrates) from the addition in quadrature of the individual reaction uncertainties. Column covers the tropospheric column.





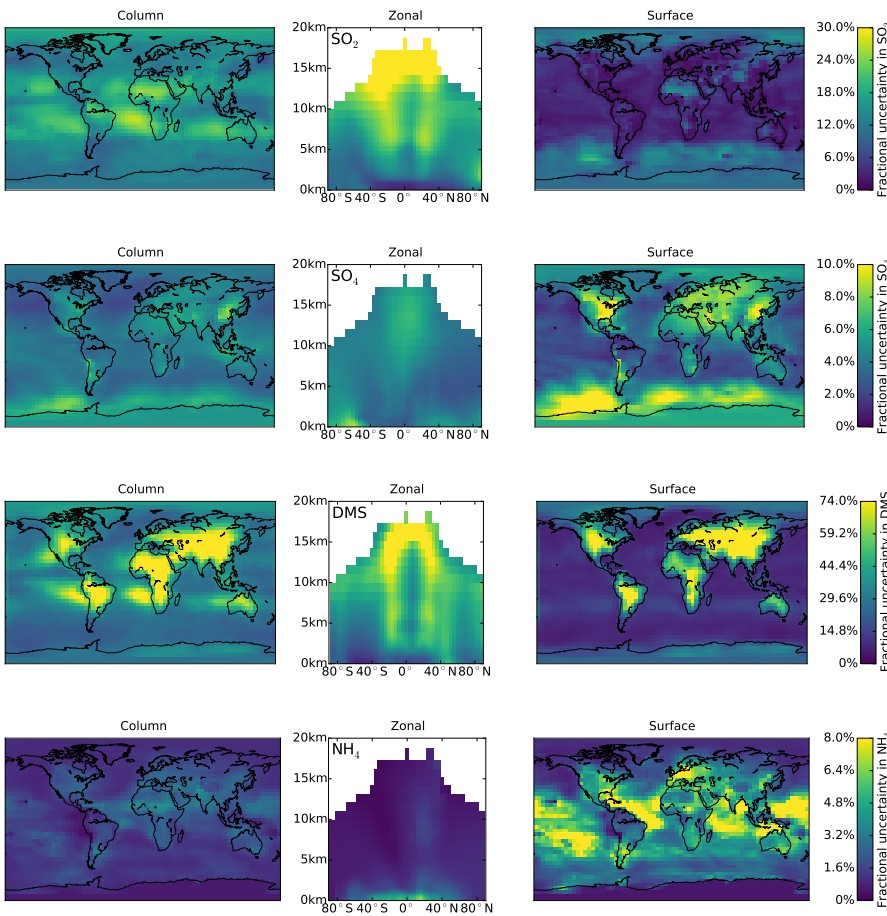

**Figure 9. Sulfur and Aerosols.** Total $1\sigma$ uncertainty in the concentrations of $SO_2$, $SO_4$, DMS (Dimethyl Sulfide) and $NH_4$ from the addition in quadrature of the individual reaction uncertainties. Column covers the tropospheric column.





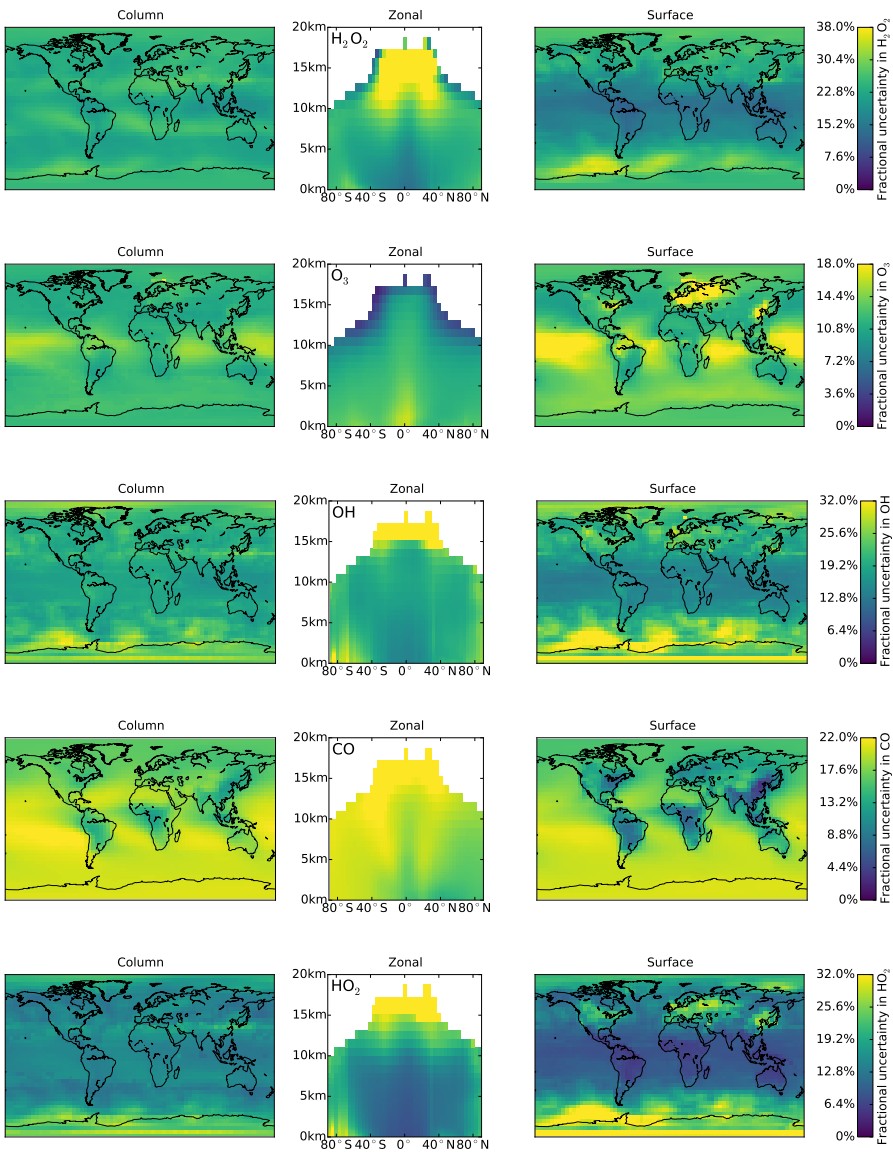

**Figure 10. Inorganics.** Total $1\sigma$ uncertainty in the concentrations of $H_2O_2$, $O_3$, OH, CO and $HO_2$ from the addition in quadrature of the individual reaction uncertainties. Column covers the tropospheric column.





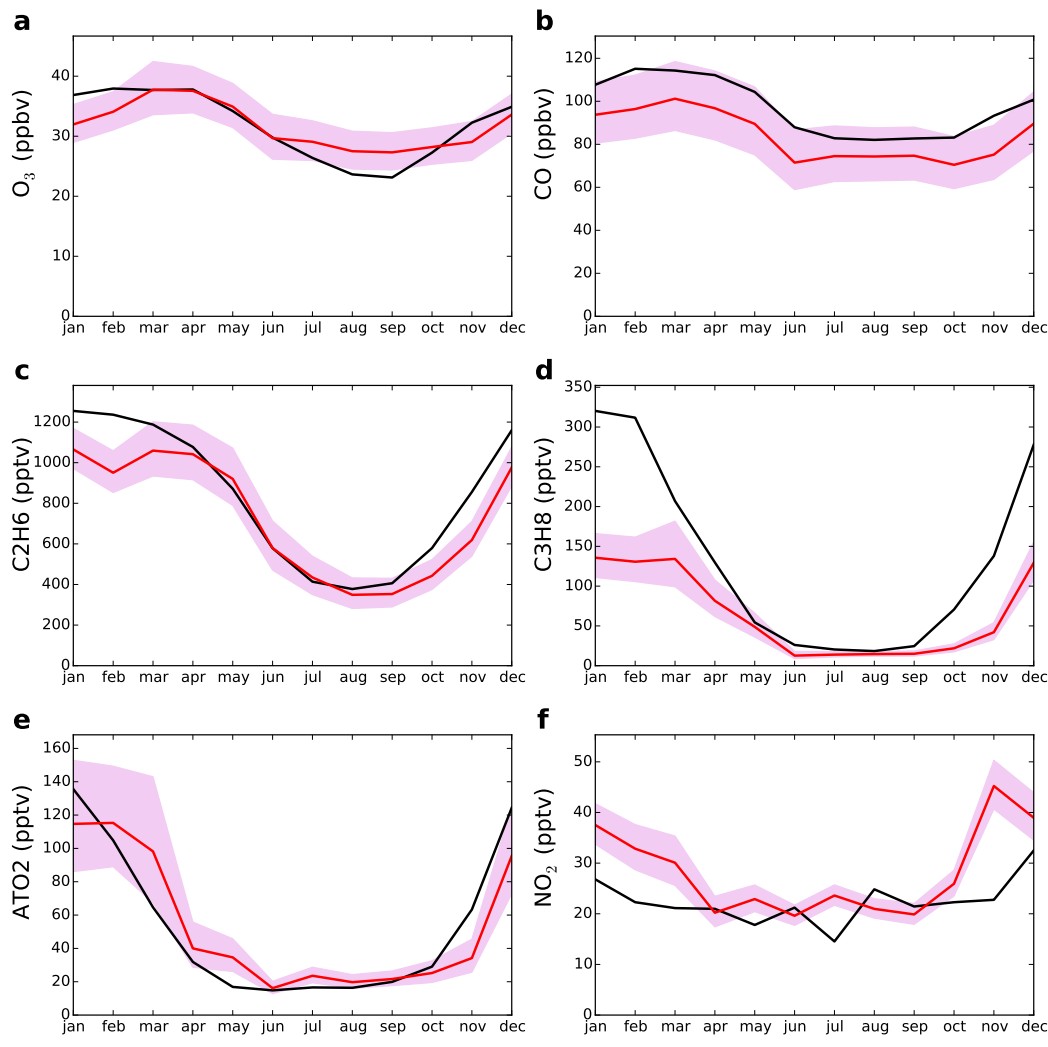

**Figure 11. Impact on model / measurement comparisons.** Modelled (red) and measured (black) annual cycle in monthly mean $O_3$, CO, $C_2H_6$, $C_3H_8$, ALK4 ($\geq$ C4 Alkanes) and $NO_2$ mixing ratios at Cape Verde (Carpenter et al., 2011). Shaded area represents the $1\sigma$ uncertainty from the 60 reactions added together in quadrature.



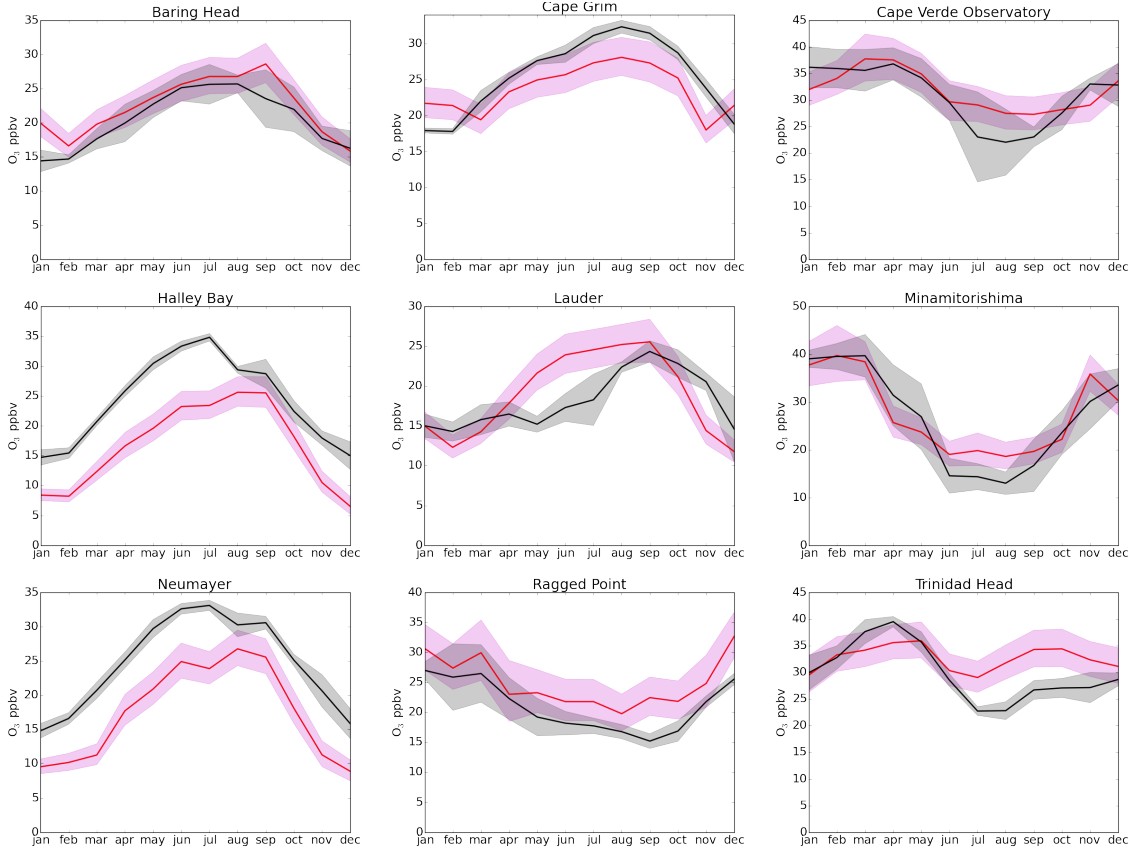

**Figure 12. Ozone site comparison** Modelled (red) and measured (black) concentrations of ozone at a range of sites. The pink shaded area shows the 1sigma uncertainty from the chemical kinetics. The error bars represent the 1sigma uncertainty of these observations. Monthly mean observational data obtained from (Sofen and Evans, 2015) (Sofen et al., 2016), using multiple years between 2004 and 2010 to create more complete datasets.





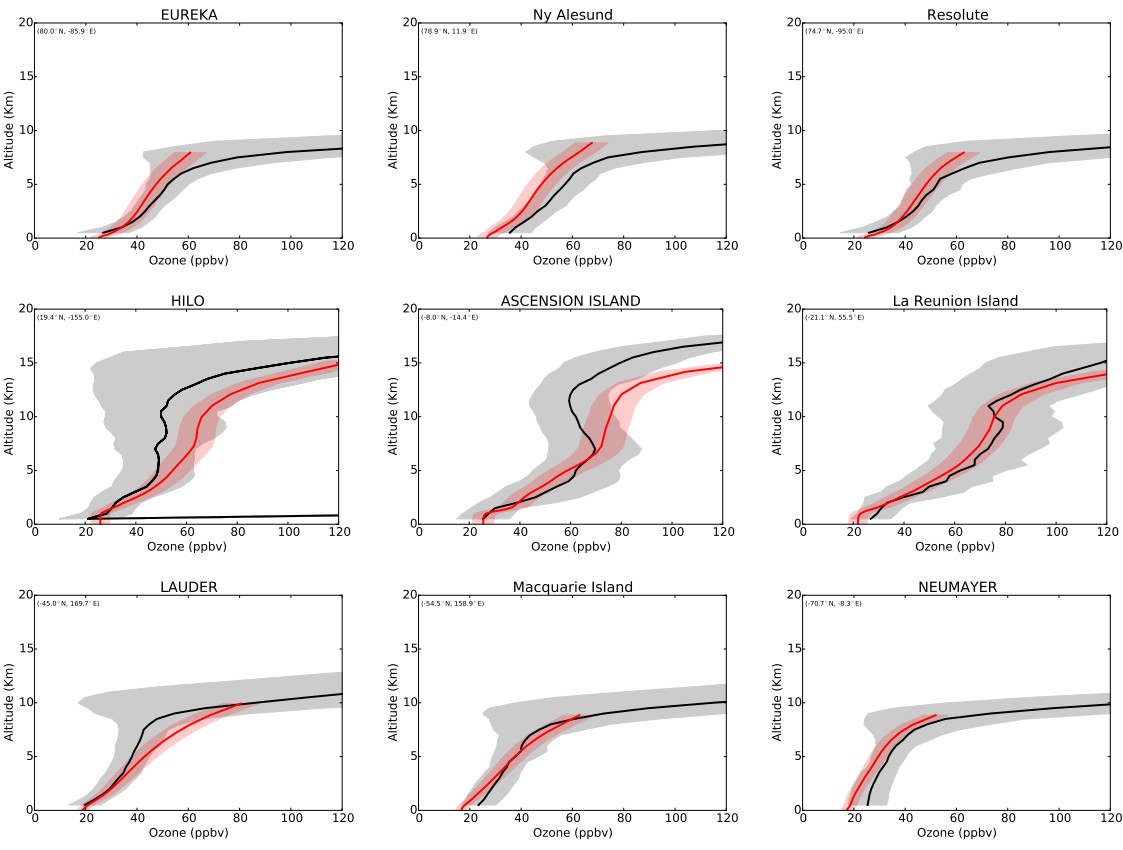

**Figure 13. Ozonesonde** Comparisons between the variability of annual ozonesonde measurements and model data with uncertainties. The black line shows the annual mean observation data and the shaded gray shows the range of data. The red line shows the model data and the pink shaded line shows the chemical 1sigma uncertainty. Observations are obtained from WOUDC (2014)(WOUDC).





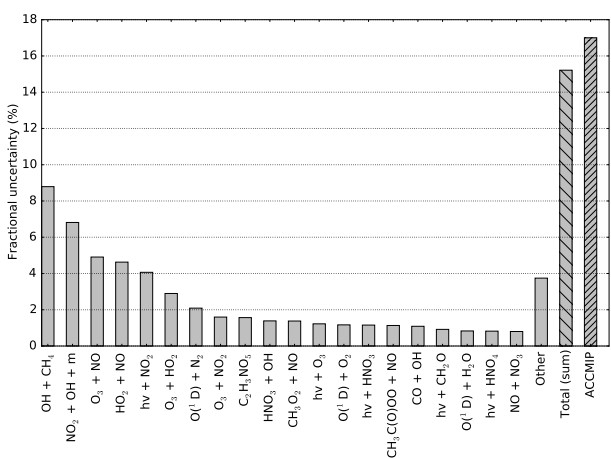

**Figure 14. Uncertainties in O$_3$ radiative forcing.** Absolute fractional uncertainty in tropospheric O$_3$ radiative forcing between the prein-dustrial and present day, due to rate constant uncertainty. Shown on the left are the 20 most important reactions. 'Other' shows the addition in quadrature of the remaining 40 reactions. 'Total (sum)' indicates the total fractional uncertainty calculated by adding together the individual uncertainties in quadrature. 'ACCMIP' indicates the inter-model spread found from the ACCMIP (Young et al., 2013) study.





**Table 1. Table of reactions studied.** f (298) indicates the JPL or IUPAC panel uncertainty estimate at 298K and g gives the rate at which this uncertainty increases away from 298K (see previous section). Reactions with 0 for the temperature dependence indicates there is zero temperature dependency or not enough information to provide a temperature varying uncertainty. The final column gives the fractional increase in the ozone burden by increasing the rate constant to its $1\sigma$ value. Reactions with a * are the 10 reactions used in the Monte Carlo study.

| Number | Reaction | f(298) | g (K) | $1\sigma$ $O_3$ burden change (%) |
|--------|----------|--------|-------|-----------------------------------|
| 1* | $NO_2 + OH \xrightarrow{M} HNO_3$ | 1.3 | 100 | -6.20 |
| 2* | $OH + CH_4 \rightarrow CH_3O_2 + H_2O$ | 1.1 | 100 | 4.15 |
| 3* | $O_3 + NO \rightarrow NO_2 + O_2$ | 1.1 | 200 | -3.61 |
| 4* | $HO_2 + NO \rightarrow NO_2 + OH$ | 1.15 | 20 | 3.09 |
| 5* | $O_3 + HO_2 \rightarrow OH + 2O_2$ | 1.15 | 80 | -2.39 |
| 6* | $O(^1D) + N_2 \rightarrow O + N_2$ | 1.1 | 20 | 1.82 |
| 7* | $O(^1D) + H_2O \rightarrow OH + OH$ | 1.08 | 20 | -1.54 |
| 8 | $HNO_3 + OH \rightarrow H_2O + NO_3$ | 1.2 | 0 | 0.928 |
| 9* | $O_3 + NO_2 \rightarrow NO_3 + O_2$ | 1.15 | 150 | -0.803 |
| 10* | $O(^1D) + O_2 \rightarrow O + O_2$ | 1.1 | 10 | 0.745 |
| 11 | $CH_3C(O)O_2 + NO \rightarrow CH_3O_2 + NO_2 + CO_2$ | 1.5 | 0 | 0.721 |
| 12* | $O_3 + OH \rightarrow HO_2 + O_2$ | 1.15 | 50 | -0.693 |
| 13 | $CO + OH \rightarrow HO_2 + CO_2$ | 1.1 | 100 | 0.571 |
| 14 | $CH_3O_2 + NO \rightarrow CH_2O + HO_2 + NO_2$ | 1.15 | 100 | 0.553 |
| 15 | $CH_3OH + OH \rightarrow HO_2 + CH_2O$ | 1.1 | 60 | 0.462 |
| 16 | $CH_3C(O)OONO_2 \rightarrow CH_3C(O)OO + NO_2$ | 1.2 | 200 | 0.341 |
| 17 | $CH_3C(O)O_2 + NO_2 \xrightarrow{M} CH_3C(O)OONO$ | 1.2 | 50 | -0.289 |
| 18 | $OH + H_2 \rightarrow H_2O + HO_2$ | 1.05 | 100 | 0.282 |
| 19 | $OH + H_2O_2 \rightarrow H_2O + HO_2$ | 1.15 | 45 | 0.265 |
| 20 | $NO + NO_3 \rightarrow 2NO_2$ | 1.3 | 100 | 0.249 |
| 21 | $HO_2 + NO_3 \rightarrow OH + NO_2$ | 1.5 | 0 | 0.248 |
| 22 | $CH_3OOH + OH \rightarrow CH_3O_2 + H_2O$ | 1.4 | 150 | -0.243 |
| 23 | $CH_3SCH_3 + OH \rightarrow SO_2 + CH_3O_2 + CH_2O$ | 1.1 | 100 | 0.231 |
| 24 | $OH + HO_2 \rightarrow H_2O + O_2$ | 1.15 | 50 | -0.215 |
| 25 | $CH_3CH_2OO + NO \rightarrow CH_3CHO + NO_2 + HO_2$ | 1.2 | 150 | 0.211 |
| 26 | $C_2H_6 + OH \rightarrow CH_3CH_2OO + H_2O$ | 1.07 | 50 | 0.201 |
| 27 | $O(^1D) + H_2 \rightarrow OH + H$ | 1.15 | 50 | 0.198 |
| 28 | $HCOOH + OH \rightarrow H_2O + CO_2 + HO_2$ | 1.2 | 100 | 0.196 |
| 29 | $OH + OH \rightarrow H_2O + O_3$ | 1.25 | 50 | 0.195 |
| 30 | $CH_3CHO + NO_3 \rightarrow HNO_3 + CH_3C(O)OO$ | 1.3 | 300 | 0.193 |





| | continued from previous page | | | |
|---|---|---|---|---|
| 31 | $HNO_2 + OH \rightarrow H_2O + NO_2$ | 1.5 | 200 | 0.178 |
| 32 | $CH_3CHO + OH \rightarrow CH_3C(O)OO + CH_2O + CO + HO_2$ | 1.05 | 20 | 0.174 |
| 33 | $CH_3SCH_3 + NO_3 \rightarrow SO_2 + HNO_3 + CH_3OO + CH_2O$ | 1.1 | 150 | 0.172 |
| 34 | $CH_3O_2 + CH_3O_2 \rightarrow CH_3OH + CH_2O + O_2$ | 1.2 | 100 | 0.170 |
| 35 | $HO_2 + HO_2 \rightarrow H_2O_2$ | 1.15 | 100 | 0.166 |
| 36 | $CH_2O + OH \rightarrow CO + HO_2 + H_2O$ | 1.15 | 50 | 0.156 |
| 37 | $NO + OH \xrightarrow{M} HNO_2$ | 1.2 | 50 | -0.151 |
| 38 | $SO_2 + OH \xrightarrow{M} SO_4 + HO_2$ | 1.1 | 100 | 0.151 |
| 39 | $NO_2 + NO_3 \xrightarrow{M} N_2O_5$ | 1.2 | 100 | -0.151 |
| 40 | $HNO_4 + OH \rightarrow H_2O + NO_2 + O_2$ | 1.3 | 500 | 0.149 |
| 41 | $OH + OH \xrightarrow{M} H_2O_2$ | 1.5 | 100 | -0.146 |
| 42 | $NO_3 + NO_3 \rightarrow 2NO_2 + O_2$ | 1.5 | 500 | -0.144 |
| 43 | $OH + NO_3 \rightarrow HO_2 + NO_2$ | 1.5 | 0 | -0.143 |
| 44 | $NO_2 + NO_3 \rightarrow NO + NO_2 + O_2$ | 2 | 0 | -0.134 |
| 45 | $HNO_4 \rightarrow HO_2 + NO_2$ | 1.3 | 100 | 0.104 |
| 46 | $HO_2 + NO_2 \xrightarrow{M} HNO_4$ | 1.1 | 50 | 0.0707 |
| 47 | $CH_3O_3 + HO_2 \rightarrow CH_3OOH + O_2$ | 1.3 | 150 | 0.0350 |
| 48 | $CH_2=C(CH_3)CH=CH_2 + OH \rightarrow HOCH_2C(OO)(CH_3)CH=CH_2$ | 1.1 | 100 | -0.0279 |
| 49 | $NO_3 + CH_2O \rightarrow HNO_3 + HO_2 + CO$ | 1.3 | 0 | -0.0145 |
| 50 | $C_4H_{10} + OH \rightarrow 2H_2O + C_4H_9$ | 1.06 | 100 | 0.0132 |
| 51 | $hv + NO_2 \rightarrow NO + O(^3P)$ | 1.1 | 0 | 2.66 |
| 52 | $hv + O_3 \rightarrow O_2 + O(^1D)$ | 1.1 | 0 | -1.97 |
| 53 | $hv + HNO_3 \rightarrow OH + NO_2$ | 1.1 | 0 | 0.559 |
| 54 | $hv + CH_2O \rightarrow CO + HO_2 + HO_2$ | 1.1 | 0 | 0.338 |
| 55 | $hv + HNO_4 \rightarrow HO_2 + NO_2$ | 1.1 | 0 | 0.262 |
| 56 | $hv + N_2O_5 \rightarrow NO_3 + NO_2$ | 1.1 | 0 | 0.223 |
| 57 | $hv + NO_3 \rightarrow NO_2 + O(^3P)$ | 1.1 | 0 | 0.222 |
| 58 | $hv + HNO_4 \rightarrow OH + NO_3$ | 1.1 | 0 | 0.200 |
| 59 | $hv + CH_3CHO \rightarrow CH_3OO + HO_2 + CO$ | 1.1 | 0 | 0.199 |
| 60 | $hv + CH_3CHO \rightarrow CH_4 + CO$ | 1.1 | 0 | 0.196 |