# Peer review of "Impact of uncertainties in inorganic chemical rate constants on tropospheric composition and ozone radiative forcing"

_Atmospheric Chemistry and Physics, 2017_

## Referee Comment (RC1) · R. Sander (Referee) · 15 Mar 2017

Newsome et al. investigate the impact of uncertainties in inorganic chemical rate constants on tropospheric composition and ozone radiative forcing. The study is well-written, very interesting and I recommend publication in ACP after considering several changes as described below.

[Figure]

**Major issues**

My only major concern is that apparently in all sensitivity studies the rate coefficients were *increased* but never *decreased*. Unless a certain reaction is the rate-limiting step inside a reaction cycle, making it faster has only a small effect on the overall rate of the cycle. However, making it slower could make this particular reaction rate-limiting and then the effect becomes large. Why was it never tested what effect a *decrease* of $k$ by 1 $\sigma$ has?

**Minor issues and technical comments**

- Abstract: "Expert panels synthesise laboratory measurements"
  Chemicals are "synthesised" but not laboratory measurements. I think it would be better to say: "Expert panels evaluate laboratory measurements".

- In the introduction you describe both the JPL and the IUPAC evaluation and then you provide Eq. (1) to decribe the uncertainty. It should be noted, however, that IUPAC does not use this definition. Instead, IUPAC defines uncertainties via $\Delta \log k$ and $\Delta E/R$. I think it would be helpful for the reader if you show how to convert between these different ways to express uncertainties.

- In your manuscript you refer to the JPL evaluation from 2011. Have you checked if the uncertainties are still the same in the more recent JPL Evaluation Number 18 from 2015?

- Page 2, line 20: Change "larger uncertanties then quoted here" to "larger uncertanties than quoted here".

- Page 3, line 2: Change "www.goes-chem.org" to "www.geos-chem.org".

- Page 7, line 5: Change "$NH_4$" to "$NH_4^+$".

- Page 9, line 7: You claim that some "reactions may appear rather un-interesting to some". What is the reason for this assumption?

- I think it would be better to call the last section "Summary" or "Conclusions" instead of "Discussion".

---

## Referee Comment (RC2) · Anonymous Referee #1 · 17 Apr 2017

The authors have used their model results to reinforce the point that uncertainties in model input kinetic parameters are sometimes significant and often greater than model to model spread. Although the results of this study come as no surprise, studies like this remind the community not to ignore the consideration of input uncertainty in comparisons of observation and model results and in directing policy decisions. The authors also point out that improved parameterizations of even extensively studied but critical reactions, such as OH + NO2 + M, would help reduce model uncertainty significantly.

This paper is well organized and presented and is suitable for publication in ACP.

* The authors have used an out of date version of the NASA/JPL data recommendations in their analysis. There does not seem to be any reasonable explanation for this oversight given in the present version of the manuscript. Although, the conclusions from the present work are likely to remain unchanged the authors should highlight any differences with the 2015 NASA/JPL data recommendations in their paper in Table 1.

* The treatment of the uncertainty in the atmospheric parameters, or lack of, is unsatisfying. A thorough treatment of photolysis uncertainty may be beyond the scope of the present work, but making an across the board percentage uncertainty assumption is surely not correct. It may have been better to not include photolysis uncertainty in the present analysis.
* * *

---

## Author Response (AR1)

**Reply to reviewers of ACPD paper: "Impact of uncertainties in inorganic chemical rate constants on tropospheric composition and ozone radiative forcing" by Ben Newsome and Mat Evans**

We thank the reviewers for their constructive comments. We address the comments below and identify changes we would make to the paper considering their comments. Where indicated we have already run the appropriate new simulations.

**Review by R Saunders.**

**Major comments:**

**My only major concern is that apparently in all sensitivity studies the rate coefficients were *increased* but never *decreased*. Unless a certain reaction is the rate-limiting step inside a reaction cycle, making it faster has only a small effect on the overall rate of the cycle. However, making it slower could make this particular reaction rate-limiting and then the effect becomes large. Why was it never tested what effect a *decrease* of k by 1 σ has?**

We had tested this previously and found that the differences were small but didn't include this in the original paper. We would suggest that we include these results in the text and show a comparison between the results of increasing and decreasing the top ten reactions to show that although there are some differences between the conclusions from increasing vs decreasing the rates the impact is small compared to the overall conclusions of the paper.

**Minor issues and technical comments**

**Abstract: "Expert panels synthesise laboratory measurements. "Chemicals are "synthesised" but not laboratory measurements. I think it would be better to say: "Expert panels evaluate laboratory measurements".**

We have updates the text to reflect this suggestion.

**In the introduction you describe both the JPL and the IUPAC evaluation and then you provide Eq. (1) to decribe the uncertainty. It should be noted, however, that IUPAC does not use this definition. Instead, IUPAC defines uncertainties via $\Delta$ log k and $\Delta$E/R. I think it would be helpful for the reader if you show how to convert between these different ways to express uncertainties.**

We have included a discussion of both methodologies in the text.

**In your manuscript you refer to the JPL evaluation from 2011. Have you checked if the uncertainties are still the same in the more recent JPL Evaluation Number 18 from 2015?**

We have updated our simulations to reflect the JPL18 evaluation and refer to that in the text. There were some minor changes which has marginally changed the order of the uncertainties of the reactions.

**Page 2, line 20: Change "larger uncertanties then quoted here" to "larger uncertanties than quoted here".**

We have corrected the text.

**Page 3, line 2: Change "www.goes-chem.org" to "www.geos-chem.org". C2**

We have corrected the text.

**Anonymous Referee #1**

**The authors have used an out of date version of the NASA/JPL data recommendations in their analysis. There does not seem to be any reasonable explanation for this oversight given in the present version of the manuscript. Although, the conclusions from the present work are likely to remain unchanged the authors should highlight any differences with the 2015 NASA/JPL data recommendations in their paper in Table 1.**

We have updated our simulations to reflect the changes made between versions 17 and 18 of the evaluation. This has made small changes to the absolute value of a couple of reactions but there is no overall change to the conclusions.

**The treatment of the uncertainty in the atmospheric parameters, or lack of, is unsatisfying. A thorough treatment of photolysis uncertainty may be beyond the scope of the present work, but making an across the board percentage uncertainty assumption is surely not correct. It may have been better to not include photolysis uncertainty in the present analysis.**

We agree that our analysis of the photolysis rates is simplistic but we believe it provides a useful context for understanding the relative role of photolysis uncertainty compared to reaction rate constant uncertainty. We think that leaving these simulations in the discussion make a useful contribution and helps to motive future work in this area. We would update the text when we are discussion this to identify the need for future improved assessments of photolysis rate uncertainties.

These dates will be inserted by Copernicus Publications during the typesetting process.

**Impact of uncertainties in inorganic chemical rate constants on tropospheric composition and ozone radiative forcing**

Ben Newsome[1] and Mat Evans[1,2]

[1]Wolfson Atmospheric Chemistry Laboratories, Department of Chemistry, University of York, York, YO10 5DD, UK.
[2]National Centre for Atmospheric Science, Department of Chemistry, University of York, York, YO10 5DD, UK.
*Correspondence to:* Mat Evans (Mat.Evans@york.ac.uk)

**Abstract.** Chemical rate constants determine the composition of the atmosphere and how this composition has changed over time. They are central to our understanding of climate change and air quality degradation. Atmospheric chemistry models, whether online or offline, box, regional or global use these rate constants. Expert panels evaluate laboratory measurements, making recommendations for the rate constants that should be used. This results in very similar or identical rate constants being used by all models. The inherent uncertainties in these recommendations are, in general, therefore ignored. We explore the impact of these uncertainties on the composition of the troposphere using the GEOS-Chem chemistry transport model. Based on the JPL and IUPAC evaluations we assess the influence of 50 mainly inorganic rate constants and 10 photolysis rates, through simulations where we increase the rate of the reactions to the 1 $\sigma$ upper value recommended by the expert panels on tropospheric composition through the use of the GEOS-Chem chemistry Transport model.

We assess the impact on 4 standard metrics: annual mean tropospheric ozone burden, surface ozone and tropospheric OH concentrations, and tropospheric methane lifetime. Uncertainty in the rate constants for $NO_2 + OH \xrightarrow{M} HNO_3$, $HO_2 + NO \rightarrow OH + NO_2$, and $O_3 + NO \rightarrow NO_2 + O_2$ are the three two largest source of uncertainty in these metrics. With the absolute magnitude of uncertainties being similar if rate constants are increased or decreased by their $\sigma$ values. We investigate two methods of assessing these uncertainties, addition in quadrature and a Monte Carlo approach, and conclude they give similar outcomes. Combining the uncertainties across the 60 reactions, gives overall uncertainties on the annual mean tropospheric ozone burden, surface ozone and tropospheric OH concentrations, and tropospheric methane lifetime of 10, 11, 16 and 16% respectively. These are larger than the spread between models in recent model inter-comparisons. Remote regions such as the tropics, poles, and upper troposphere are most uncertain. This chemical uncertainty is sufficiently large to suggest that rate constant uncertainty should be considered alongside other processes when model results disagree with measurement.

Calculations for the pre-industrial allow a tropospheric ozone radiative forcing to be calculated of 0.412 $\pm$ 0.056 0.062 Wm$^{-2}$. This uncertainty (14 13 %) is comparable to the inter-model spread in ozone radiative forcing found in previous model-model inter-comparison studies where the rate constants used in the models are all identical or very similar. Thus the uncertainty of tropospheric ozone radiative forcing should expanded to include this additional source of uncertainty. These rate constant uncertainties are significant and suggest that refinement of supposedly well known chemical rate constants should be considered alongside other improvements to enhance our understanding of atmospheric processes.

**1 Introduction**

The concentration of gases and aerosols in the atmosphere have changed over the last century due to human activity. This has resulted in a change in climate (**?**) and a degradation in air quality (**?**) with tropospheric ozone ($O_3$) and methane ($CH_4$) playing a central role. The response of these compounds to the changing emissions is complex and non-linear (**?**). The hydroxyl radical (OH) plays a central role in this chemistry as it initiates the destruction of many pollutants (notably $CH_4$) and so determines their lifetime in the atmosphere. The dominant source of OH is the photolysis of $O_3$ in the presence of water vapour. The oxidation of compounds such as $CH_4$, carbon monoxide (CO) and other hydrocarbons can lead to the production of $O_3$ if sufficient oxides of nitrogen ($NO_x$) are present. Changes in the emissions of $O_3$ precursors between the pre-industrial ($\sim$1850) and the present day have increased $O_3$ concentrations and this has produced a radiative forcing estimated to be $410 \pm 65$ mWm$^{-2}$ (**?**).

The rate constants of the reactions occurring in the atmosphere have been determined by a number of laboratory studies which are  synthesised by groups such as the IUPAC (**?**) and JPL (**?**) panels. These provide recommendations for both rate constants and their associated uncertainties. These reactions are typically expressed in an Arrhenius form to represent the temperature dependence. More complicated representations are needed for three-body reactions.  IUPAC and JPL provide similar but different representations of the uncertainty in  a rate constant. For IUPAC (Eqn. 2) the uncertainty in a rate constant is described as the uncertainty in the $\log_{10}$ of the rate constant ($\Delta \log_{10} k_T$) at a temperature (T), with the panel giving values for the $\log_{10}$ uncertainty at 298K ($\Delta \log_{10} k_{298K}$) and the rate of increase in uncertainty away from 298K described by a $\Delta E/R$ term. For JPL (Eqn. 2) the relative uncertainty in a rate constant (f(T)) is described as the relative uncertainty at temperature of 298K (f(298)) together with a term (g) that expresses how quickly the uncertainty increases away from 298K (Equation 1), leading to temperature dependences which increase away from room temperature (Figure 1).

$$f(\mathrm{T}) = f(298\mathrm{K}) \exp \left| g \left( \frac{1}{\mathrm{T}} - \frac{1}{298\mathrm{K}} \right) \right|$$

$$\Delta \underset{\sim}{\log_{10} k(T)} \log_{10} k_T = \Delta \underset{\sim}{\log_{10} k(298K)} \log_{10} k_{298K} + 0.4343 \frac{\Delta E}{R} \left( \frac{1}{T} \frac{1}{\mathrm{T}} - \frac{1}{298} \frac{1}{298\mathrm{K}} \right) \tag{1}$$

[revised manuscript text omitted]

We thank the reviewers for their constructive comments. We address the comments below and identify changes we would make to the paper considering their comments. Where indicated we have already run the appropriate new simulations.

**Review by R Saunders.**

**Major comments:**

**My only major concern is that apparently in all sensitivity studies the rate coefficients were *increased* but never *decreased*. Unless a certain reaction is the rate-limiting step inside a reaction cycle, making it faster has only a small effect on the overall rate of the cycle. However, making it slower could make this particular reaction rate-limiting and then the effect becomes large. Why was it never tested what effect a *decrease* of k by 1 σ has?**

We had tested this previously and found that the differences were small but didn't include this in the original paper. We would suggest that we include these results in the text and show a comparison between the results of increasing and decreasing the top ten reactions to show that although there are some differences between the conclusions from increasing vs decreasing the rates the impact is small compared to the overall conclusions of the paper.

**Minor issues and technical comments**

**Abstract: "Expert panels synthesise laboratory measurements. "Chemicals are "synthesised" but not laboratory measurements. I think it would be better to say: "Expert panels evaluate laboratory measurements".**

We have updates the text to reflect this suggestion.

**In the introduction you describe both the JPL and the IUPAC evaluation and then you provide Eq. (1) to decribe the uncertainty. It should be noted, however, that IUPAC does not use this definition. Instead, IUPAC defines uncertainties via $\Delta \log k$ and $\Delta E/R$. I think it would be helpful for the reader if you show how to convert between these different ways to express uncertainties.**

We have included a discussion of both methodologies in the text.

**In your manuscript you refer to the JPL evaluation from 2011. Have you checked if the uncertainties are still the same in the more recent JPL Evaluation Number 18 from 2015?**

We have updated our simulations to reflect the JPL18 evaluation and refer to that in the text. There were some minor changes which has marginally changed the order of the uncertainties of the reactions.

**Page 2, line 20: Change "larger uncertanties then quoted here" to "larger uncertanties than quoted here".**

We have corrected the text.

**Page 3, line 2: Change "www.goes-chem.org" to "www.geos-chem.org". C2**

We have corrected the text.

**Anonymous Referee #1**

**The authors have used an out of date version of the NASA/JPL data recommendations in their analysis. There does not seem to be any reasonable explanation for this oversight given in the present version of the manuscript. Although, the conclusions from the present work are likely to remain unchanged the authors should highlight any differences with the 2015 NASA/JPL data recommendations in their paper in Table 1.**

We have updated our simulations to reflect the changes made between versions 17 and 18 of the evaluation. This has made small changes to the absolute value of a couple of reactions but there is no overall change to the conclusions.

**The treatment of the uncertainty in the atmospheric parameters, or lack of, is unsatisfying. A thorough treatment of photolysis uncertainty may be beyond the scope of the present work, but making an across the board percentage uncertainty assumption is surely not correct. It may have been better to not include photolysis uncertainty in the present analysis.**

We agree that our analysis of the photolysis rates is simplistic but we believe it provides a useful context for understanding the relative role of photolysis uncertainty compared to reaction rate constant uncertainty. We think that leaving these simulations in the discussion make a useful contribution and helps to motive future work in this area. We would update the text when we are discussion this to identify the need for future improved assessments of photolysis rate uncertainties.

These dates will be inserted by Copernicus Publications during the typesetting process.

**Impact of uncertainties in inorganic chemical rate constants on tropospheric composition and ozone radiative forcing**

Ben Newsome[1] and Mat Evans[1,2]

[1]Wolfson Atmospheric Chemistry Laboratories, Department of Chemistry, University of York, York, YO10 5DD, UK.
[2]National Centre for Atmospheric Science, Department of Chemistry, University of York, York, YO10 5DD, UK.

*Correspondence to:* Mat Evans (Mat.Evans@york.ac.uk)

**Abstract.** Chemical rate constants determine the composition of the atmosphere and how this composition has changed over time. They are central to our understanding of climate change and air quality degradation. Atmospheric chemistry models, whether online or offline, box, regional or global use these rate constants. Expert panels evaluate laboratory measurements, making recommendations for the rate constants that should be used. This results in very similar or identical rate constants being used by all models. The inherent uncertainties in these recommendations are, in general, therefore ignored. We explore the impact of these uncertainties on the composition of the troposphere using the GEOS-Chem chemistry transport model. Based on the JPL and IUPAC evaluations we assess the influence of 50 mainly inorganic rate constants and 10 photolysis rates, on tropospheric composition through the use of the GEOS-Chem chemistry Transport model.

We assess the impact on 4 standard metrics: annual mean tropospheric ozone burden, surface ozone and tropospheric OH concentrations, and tropospheric methane lifetime. Uncertainty in the rate constants for $NO_2 + OH \xrightarrow{M} HNO_3$,  and $O_3 + NO \rightarrow NO_2 + O_2$ are the  two largest source of uncertainty in these metrics. With the absolute magnitude of uncertainties being similar if rate constants are increased or decreased by their $\sigma$ values. We investigate two methods of assessing these uncertainties, addition in quadrature and a Monte Carlo approach, and conclude they give similar outcomes. Combining the uncertainties across the 60 reactions, gives overall uncertainties on the annual mean tropospheric ozone burden, surface ozone and tropospheric OH concentrations, and tropospheric methane lifetime of 10, 11, 16 and 16% respectively. These are larger than the spread between models in recent model inter-comparisons. Remote regions such as the tropics, poles, and upper troposphere are most uncertain. This chemical uncertainty is sufficiently large to suggest that rate constant uncertainty should be considered alongside other processes when model results disagree with measurement.

Calculations for the pre-industrial allow a tropospheric ozone radiative forcing to be calculated of $0.412 \pm$  0.062 $Wm^{-2}$. This uncertainty ( 13 %) is comparable to the inter-model spread in ozone radiative forcing found in previous model-model inter-comparison studies where the rate constants used in the models are all identical or very similar. Thus the uncertainty of tropospheric ozone radiative forcing should expanded to include this additional source of uncertainty. These rate constant uncertainties are significant and suggest that refinement of supposedly well known chemical rate constants should be considered alongside other improvements to enhance our understanding of atmospheric processes.

**1  Introduction**

The concentration of gases and aerosols in the atmosphere have changed over the last century due to human activity. This has resulted in a change in climate (**?**) and a degradation in air quality (**?**) with tropospheric ozone ($O_3$) and methane ($CH_4$) playing a central role. The response of these compounds to the changing emissions is complex and non-linear (**?**). The hydroxyl radical (OH) plays a central role in this chemistry as it initiates the destruction of many pollutants (notably $CH_4$) and so determines their lifetime in the atmosphere. The dominant source of OH is the photolysis of $O_3$ in the presence of water vapour. The oxidation of compounds such as $CH_4$, carbon monoxide (CO) and other hydrocarbons can lead to the production of $O_3$ if sufficient oxides of nitrogen ($NO_x$) are present. Changes in the emissions of $O_3$ precursors between the pre-industrial ($\sim$1850) and the present day have increased $O_3$ concentrations and this has produced a radiative forcing estimated to be $410 \pm 65$ $mWm^{-2}$ (**?**).

The rate constants of the reactions occurring in the atmosphere have been determined by a number of laboratory studies which are  synthesised by groups such as the IUPAC (**?**) and JPL (**?**) panels. These provide recommendations for both rate constants and their associated uncertainties. These reactions are typically expressed in an Arrhenius form to represent the temperature dependence. More complicated representations are needed for three-body reactions.  IUPAC and JPL provide similar but different representations of the uncertainty in  a rate constant. For IUPAC (Eqn. 2) the uncertainty in a rate constant is described as the uncertainty in the $\log_{10}$ of the rate constant ($\Delta \log_{10} k_T$) at a temperature (T), with the panel giving values for the $\log_{10}$ uncertainty at 298K ($\Delta \log_{10} k_{298K}$) and the rate of increase in uncertainty away from 298K described by a $\Delta E /R$ term. For JPL (Eqn. 2) the relative uncertainty in a rate constant (f(T)) is described as the relative uncertainty at temperature of 298K (f (298)) together with a term (g) that expresses how quickly the uncertainty increases away from 298K (Equation 1), leading to temperature dependences which increase away from room temperature (Figure 1).

$$f(\text{T}) = f(298\text{K}) \exp\left| g\left(\frac{1}{\text{T}} - \frac{1}{298\text{K}}\right)\right|$$

$$\Delta \log_{10} k_T = \Delta \log_{10} k_{298K} + 0.4343\frac{\Delta E}{R}\left(\frac{1}{T} - \frac{1}{298K}\right) \tag{1}$$

[revised manuscript text omitted]